# Friends in Sports: Social Networks in Leisure, School and Social Media

**DOI:** 10.3390/ijerph18126197

**Published:** 2021-06-08

**Authors:** Håvard Bergesen Dalen, Ørnulf Seippel

**Affiliations:** Department of Sport and Social Sciences, Norwegian School of Sport Sciences, 0863 Oslo, Norway; ornulf.seippel@nih.no

**Keywords:** social networks, friends, youth, sports, school, leisure, social media

## Abstract

Young athletes value their social relations in sports, and these social relations can have consequences when it comes to joining, continuing, and quitting sports. Yet the important question of how social relations in sports develop has not yet been adequately answered. Hence, we investigated how athletes’ social relations in sports depend on social relations outside of sports: in leisure, school, and social media. A total of 387 athletes (aged 16–19) from 30 Norwegian sports groups completed a survey on electronic tablets. We asked how social relations in leisure, school, and social media—through the social mechanisms of contact, homophily, and contagion—influenced social relations in sports. We also controlled for the effect of exercise frequency and duration (years) of contact in sports. Exponential random graph modelling (ERGM) analyses showed that first and foremost, relations from social media and leisure, but also school networks and exercise frequency, influence sports networks. This study shows that social relations in sports are diverse and depend on social relations outside sports. We discuss how this has ‘counterintuitive’ consequences for sports participation, particularly the importance of supporting athletes’ social relations outside of sports for the strengthening of social relations within sports when addressing challenges concerning recruitment, continuation, and dropout from sports.

## 1. Introduction

Both quantitatively and qualitatively, sports are among young people’s most important social arenas [1,2,3]. In trying to understand the meaning of young athletes’ sports participation, a substantial amount of research shows that the social relations and social experiences young people have in sports are among their main reasons for taking part in sports [4,5]. There are also several studies on what these social relations look like [6,7] and the consequences of the qualities of social relations in sports. Research shows that social relations matter for how sports are experienced (fun and enjoyment) [8], feelings of belonging and integration [9,10,11], social capital [12,13,14,15], health and lifestyle [16,17] and the levels of participation in and dropping out from sports [4,18,19,20]. Yet, commenting on the situation of social network studies in general, Small claims that ‘… in their devotion to studying the consequences of social ties, many researchers have taken for granted the process from which ties arise’ ([21], p. 8). We argue that this observation is also valid for sports studies, and accordingly, in the current study, we investigate the consequential issue of how the often-praised social relations in youth sports develop.

Studies from various social and organizational fields have shown that recruitment to, participation in, and attrition from organized activities are not primarily about individual characteristics and motives but are first and foremost about social networks and opportunities for action. The existence of conducive, vibrant, and inviting social networks are prerequisites to organizational life [22,23]. To start or continue in sports, there should be an opportunity to meet other people to do sports with, and there should be some kind of (continued) social encouragement to keep utilizing this opportunity: ‘… networks do not arise out of thin air. People’s networks emerge over the course of their routine activities, in the everyday organizations where those activities take place’. Hence, in the current study, we adopt a holistic view of young athletes’ social lives and ask how the quantity and quality of young athletes’ social relations in sports depend on participation in social arenas outside sports.

The present study contributes to previous research in two ways. First, we describe young people’s social relations in sports as social networks. Although several scholars have argued for the usefulness of studying social networks in the field of sports [6,24,25], there is still a dearth of network research on the social aspects of sports groups [7]. Second, and more importantly, we investigate how young people’s social relations in sports develop and how the development of athletes’ ties are connected to their social ties outside of sports.

To guide our analyses and interpret our findings, we present a theoretical framework built on three pillars. Based on the philosophy of sports, we first show how sports have inherent qualities conducive to friendship and the development of social relations. We subsequently supplement these insights with three social mechanisms from the social network literature: contact, homophily, and contagion [26]. To grasp the social context of sports, we focus on three social arenas that are key to most young people’s lives: school, non-sport leisure, and social media. Third, considering the social characteristics of these arenas and the three social network mechanisms, we present a set of hypotheses for how participation in non-sports networks might matter for the development of social ties in sports.

To answer the question of how the youth’s social relations in sports are influenced by social participation in other arenas, we start with an outline of the theoretical framework and previous research. Next, we present the data and methods. The results section has two parts. First, we present some basic statistics on what athletes’ sports networks look like (degrees, density, centralization) and the extent to which they overlap (how many of those in sports that also share non-sport relations: school, leisure, and social media). Second, with the help of exponential random graph modelling (ERGM) analyses, we show how the strengths of social relations in youth sports depend on participation in non-sports networks. We conclude the article by providing a summary of our empirical findings and interpreting them more thoroughly in light of the theoretical framework. We also discuss how our findings have implications for some very much discussed topics in sports research: recruitment to, continuation in, and dropout from organized youth sports.

## 2. Theories, Contexts, and Previous Research

### 2.1. Sports’ Inherent Social Potential

A common belief is that sports promote friendship and social relations. Jones [27] states that “…, sport seems to me to be especially conducive to friendship” ([27], p. 131). Why are sports valuable for social relations? A first and obvious answer is that participation in sports sustains interactions—physically and socially—at one place and at one time. Second, sports connect people with similar interests in a collective effort; they work together while doing something they care about. Next, the commonness and future-oriented nature of sports could, when fulfilled, pave the way for strong social relations. Sports have the potential to transcend ordinary everyday interactions and unite people in social experiences favorable to social relations. Accounts of such experiences have been conceptualized as flow [28], aesthetics [29], or religious experiences [30,31]. Hence, a basic assumption for our study is that sports provide fertile ground for social relations.

However, not all social relations in sports can have the elevated character described in this philosophical theory. The sociology of friendships has found that most individuals have a layer of social relations with only a few core ties (two to five people), a wider set of sympathy social relationships (15–17 people), and an even larger extended social network of around 150 persons [32,33]. That 93% of Norwegian youth take part in sports for shorter or longer periods also indicates that their social relations in sports should be diverse [1]. The high number of dropouts also points to variations in attachment to sports [19].

To grasp some of this diversity in social relations, we build on the seminal distinction between weak and strong networks [34]. On the one hand, sports prepare people for a type of close social relations, which we label strong sports networks. We assume that these relations are intimate, committed, and demanding. On the other hand, our discussions point to the prevalence of more superficial and less intimate and less demanding social relations within sports. We label these relations weak sports networks. As argued in the theories of “the strength of weak ties” [34], it is not that these weaker networks are necessarily less consequential—they might have important social functions and are, for our purposes, essentially different from the strong sports networks. Therefore, on top of a basic propensity for social relations in sports, we also assume that there are both tighter and looser social relations among athletes.

### 2.2. Social Network Analysis: Three Social Mechanisms

The purpose of our analyses is to sort out what matters in the probability of developing social relations in sports. To better address these queries, three social mechanisms from social network studies are pertinent.

First, contact theory states that people have to meet physically in space and time to develop social networks [35]. Sports are considered to provide a social environment that is conducive to such contact opportunities and is potentially a versatile place for developing social relations. Contact matters both for the development of relations within sports and for the way social relations outside sports influence social relations within sports. Contact theory also shows how we should expect social relations outside sports to influence social relations within sports: social relations in sports could be strengthened because athletes also interact in other arenas. Previous research supports this assumption, showing how coattending different social activities tends to strengthen friendships [36,37,38].

Contagion is a social mechanism describing the processes where exposure to resources flowing through networks—knowledge, emotions, goods, money, and so forth—influences human knowledge, attitudes, and behavior. In this way, contagion indicates that people meeting through certain networks will become more similar to each other [39]. For our study, contagion implies that athletes who spend time together outside sports in a non-sport network will tend to become more like each other, thereby potentially developing their social relations in sports. We also assume that some social arenas are more contagious than others because the interactions in these arenas have qualities that are more (or less) conducive to social relations [40]. We will return to contagion effects when presenting the social arenas.

Whereas contagion points to how influence occurs in social processes, a third network mechanism involves a selection effect: homophily. The idea is that people with similar characteristics, interests, and experiences attract each other and tend to establish social relations [41]. In our case, we expect two types of effects of the homophily mechanism. First, similar people, regardless of having met previously, will be attracted to each other when they meet in sports. Second, people who have participated together in one social arena will tend to seek each other out in new social arenas because they are similar regarding this shared previous experience. In short, the effect of homophily in sports will depend on having another arena as a common reference, or more consequentially, having common experiences in other arenas. We will specify our expectations when we discuss the particularities of our chosen social arenas in the following sections.

### 2.3. Social Arena Mechanism: Voluntariness and Exclusiveness

In this section, we describe the social arenas included in the study and discuss how the characteristics of social relations in these arenas have implications for the development of social relations in sports.

*Sports*. The sports clubs in our study are voluntary organizations and part of the Norwegian Olympic and Paralympic Committee and Confederation of Sports (NIF), an umbrella organization that organizes 55 national sports federations, 19 regional sports federations, and approximately 12,000 local sports clubs. Even though the NIF is partly funded by public resources, voluntary work is the most important resource for most Norwegian sports clubs.

The fact that youth sports themselves are voluntary as well as part of the voluntary sector has consequences for social relations in sports [42]. Being affiliated with sports clubs is voluntary, whereas participation in most other social arenas (family, school, work, etc.) is compulsory (or at least, less voluntary). Hirschman’s [43] theory on exit, voice, and loyalty captures a social mechanism that is relevant for this voluntary–compulsory distinction. When one is free to join and exit an organization, staying with the organization implies a certain commitment to the group and an obligation towards co-members. Otherwise, one would leave. This is an argument in favor of the idea that social relations in sports are more committed than social relations in less voluntary settings.

For Norwegian sports, the dominant policy aim is “sports for all” [44], which builds on a vision to create and sustain an inclusive social environment with equal opportunities for all young people to participate in sports. At first glance, empirical research indicates that these policies are successful, and accordingly, that being a member of sports clubs is not very distinctive or exclusive. Dropping out from sports, however, occurs at a high rate when the athletes reach the age of our respondents (16–19) [45], so remaining affiliated with sports at this age would at least reflect a certain devotion and dedication to sports. As a second social-arena-specific mechanism, we suggest that the more exclusive a social arena is, the more conducive it is to the development of social relations.

*Leisure* consists of a broad range of formal and informal free-time activities—from the highly organized (as sports) to the very free activity of just meeting friends regularly at or outside home. Recent research also shows that this is an important social arena for most young people, both quantitatively [46] and because it has qualitative implications for young people’s lives in general [47]. We assume that social interactions during one’s free time is voluntary, and because most of these activities are less prevalent and carried out in smaller groups than sports (as well as school and social media), they are also more exclusive. This implies that on average, leisure activities represent social arenas conducive to building social relations. For the three social network mechanisms, leisure activities provide—although to varying degrees—a good deal of contact, they build on homophily (people show up to do what they like with others who like the same activities), and they are contagious. As such, many leisure activities are helpful from a social network perspective when it comes to developing social relations.

*Schools*. Recent figures show that 97% of Norwegian youth enroll in upper secondary school the same year they complete compulsory education [48]. As such, for our respondents, school is not voluntary and not very exclusive; therefore, school theoretically plays a relatively weak role as a provider of stronger social relations. A fundamental difference between the roles of local school and local sports clubs is that school is compulsory, whereas sports are coupled with freely chosen activities [49].

For the social network mechanisms, schools provide high levels of contact, which support contagion: young people’s continuous interactions over the years should contribute to social relations. The homophily mechanisms are probably relatively weak, especially compared with sports and leisure, where exclusivity makes for more similarly motived participants.

However, two factors suggest a more positive social role for schools. First, school life is important because of the quantity of time spent there and the consequences of school results on one’s success later on in life [50]. Second, it is also the case that going to school often implies a type of identity marker. Hence, even though schools as social arenas lack some of the qualities that make them socially significant—almost compulsory and non-exclusive—there are also clear indications that they could be conducive to strong social relations.

*Social media*. In today’s network society, the use of social media for connecting with others has exploded, and for many young people, it is a massive and time-consuming part of their everyday lives [51]. Close to all Norwegian youth in our targeted age group use social media for instant messaging, putting them at the top end in Europe when it comes to social media use [52]. Especially popular are instant messaging apps (e.g., Snapchat) designed for smartphones, which are more exclusive than, for example, Facebook. Because ‘everyone’ is (always) online and easily accessible, social media interaction has the potential to influence (i.e., strengthen or weaken) social relationships, including relationships outside social media. Social media is voluntary. However, social media is also inclusive, with a low threshold for participation. In sum, we assume that these characteristics imply low levels of loyalty to interactions in this arena; the exit logic does not really apply to social media. One could easily stay on without strong social commitments to others in this arena. Although the social mechanisms of contact, contagion, and homophily have a certain relevance for social media interactions, the effects of such mechanisms are—because of the virtual character of interactions—probably weaker than in real life interactions. Thus, social media lacks exclusivity, and because of its lack of face-to-face interactions, it probably involves a high volume of low-intensity social bonds.

In short, we assume that sports have a high level of social potential. The organizational structure of sports—as a voluntary activity in voluntary organizations—also adds to the potential for such social qualities. On top of this baseline, we have outlined two sets of social mechanisms that indicate how social relations in sports depend on the social ties stemming from elsewhere. From the social network theory, we can see how social relations depend on and work through contact, contagion, and homophily. In our description of the social arenas included in the current study, we have shown how their voluntariness and exclusiveness prepare for different social relations and effects.

All the social relations we study reflect these social mechanisms to a certain degree, but they do so differently. We hypothesize that more frequent contacts in sports (H1: Frequency) and more durable contacts (in years) (H2: Affiliation) will lead to more social relations in sports. The qualities of the social relations in the leisure, school, and social media arenas differ in many ways, and we hypothesize that leisure is the most intense and exclusive social arena, having the strongest outside effect on social relations in sports (H3: Leisure). We further assume that social relations in social media have a stronger effect on sports’ social life than school, which is the least voluntary and exclusive, yet we also approach the social media effect as a more open question (H4: Social Media). An important part of our study aims to show that not all social relations in sports are necessarily deep, intimate, or committed, and we assume that what we call strong social networks depends on non-sport social relations more than weak sports networks (H5: Weak vs. strong networks).

## 3. Materials and Methods

*Data.* We surveyed the social relations of 387 young athletes in 30 groups in sports clubs. Examples of groups are girls aged 16 playing handball in a club, boys aged 17 playing football, and an age group (often wider, e.g., 16–18 years) participating in cross country skiing.

The data collection started by contacting coaches from the first author’s personal network, generally by phone. The coaches were informed about the aim of the project and were asked whether they and their team wanted to participate. We sent the accepting coaches a description of the research project and asked them to return a list of the athletes who wanted to participate. The coaches informed their athletes that participation was voluntary.

We surveyed the respondents on electronic tablets immediately after training sessions or social gatherings. Completion of the questionnaire took about 20 min. Absent athletes received the survey by email, followed by a reminder if the survey was not completed within one week. We registered respondents as missing if they had not completed the survey after three reminders. The final response rate was 74% (387 of the 518 athletes who consented to participate). The response rate (at the team level) varied between 37% and 100%. The average team size was 12.9 (min. 6, max. 20, SD = 3.4). The final sample consisted of 46% girls (56% boys), with an average age of 17.1 years (SD = 1.5). The athletes belonged to 8 ski groups, 11 football groups, and 11 handball groups from 8 out of the 18 Norwegian counties. With respect to gender, 11 groups were exclusively boys, 11 were exclusively girls, and the 8 ski groups were all mixed gender. All ethical aspects of the study were approved by the Norwegian Centre for Research Data (NSD).

*Measures*. The weak sports network includes not very demanding and non-intimate social interactions, and we asked the respondents to select others they felt comfortable being with in everyday interactions: ‘Who do you usually talk to during breaks in practice sessions?’ We operationalized the strong sports network by asking the following: ‘With which members of the group do you usually share a hotel room or sleep next to during away games or competitions?’ This question points to close, intimate, and trustful relations. The school network captures social relations in the school context: ‘Which team members attend or have attended the same school as you?’ We mapped social media networks by asking the following: ‘Who do you usually send pictures or video snippets to (e.g., with Snapchat)?’ Leisure is a wide category, and young people vary in how they spend their free time. Accordingly, we included a broad range of activities and asked, ‘Over the last two weeks, with whom of your team members have you done the following activities?’ The respondents answered this question by selecting from a list the co-athletes with whom they had ‘been shopping’; ‘seen sports, either live or on the TV’; ‘been out eating’; ‘been skateboarding, snowboarding, or taking part in other non-organized activities’; ‘played computer/TV games’; ‘visited at [co-athlete’s] home’; ‘[co-athlete] visited me at my house’; ‘hung out without doing anything in particular (e.g., been outside, at the mall)’; ‘been hiking’; and ‘visited the movies or theatre with’. We then used these measures to construct an index consisting of a matrix with a binary structure indicating whether the actors had met in one way or another. All networks are directed.

*Analyses*. We describe the strong and weak networks by measuring average degree, density, and degree centralization. Average degree counts the average number of social relations a member has on each team ([53], p. 181). Density is the number of social relations in the network divided by the number of possible social relations, which informs us about how connected the networks are ([53], p. 181). Centralization summarizes the distribution of relationships in the groups and functions as a measure of hierarchical structures, that is, whether some members have more relationships than others ([53], p. 180).

The data were analyzed using ERGM, which models each of the sports networks (weak and strong) as a function of their members’ participation in non-sports networks. The method estimates the probability that sports team members develop social relations with their co-athletes, here taking into account the group members’ basic propensity to establish social relations, the intensity and duration of their sports participation, and their participation in non-sports networks [54].

Our networks are binary; therefore, the interpretation of ERGM models is much like a logistic regression, with the main difference being that the unit of analysis is the ties between nodes (and not individual attributes). Thus, coefficients are the change in the log-odds’ likelihood of a tie for a unit change in predictor.

In some of the groups, the school, leisure, and social media networks perfectly predicted the ties in the sports networks in the logistic ERGM regression models (e.g., all members of a sports network went to the same school). This is known as “separation” and causes maximum likelihood estimations to produce implausible results ([55], pp. 88–90). We handled this problem by adding a penalty term that shrank unrealistic values (the values furthest away from zero) from the maximum likelihood estimation towards zero [56]. The penalty term reduces bias and yields interpretable effect sizes. The drawback is that standard errors must be interpreted with caution because they arise from a bias deliberately placed on the maximum likelihood estimation. For the ERGM analyses of strong networks, we excluded three teams that had too few respondents and/or relations for the ERGM models to produce interpretable results (model degeneracy) [54]. We used R [57] and the statnet package to analyze our data [58].

To control for model fit, we compared the Akaike’s Information Criterion (AIC) values of our models to the null-models. Though there are no specific cutoffs for AIC, a smaller AIC-value signifies a better fit [59]. A total of 88% of the ERGM models (i.e., 51 of 57 models) with the chosen independent variables had smaller AIC values than the simple models, providing support for the chosen model.

## 4. Results

We studied social relations in 30 sports groups (Table 1). For each team, we investigated five types of relations between the athletes in the team, and we categorized these five networks as ‘strong sports networks’, ‘weak sports networks’, ‘school networks’, ‘leisure networks’, and ‘social media networks’. Athletes with strong sports relations to their co-athletes are part of strong sports networks. Athletes with weak sports relations to their co-athletes are part of weak sports networks. In addition, we describe three sets of relations among the athletes in each team based on their relations to each other outside sports. Those within each team going to the same school belong to what we call a school network, teammates who share a leisure activity are part of a leisure network, and those athletes who also meet on social media constitute a social media network. Most athletes have relations of different types and qualities in relation to their co-athletes, so each athlete could be part of more than one network. For example, one athlete could have weak relations to ten of their co-athletes, strong relations to two co-athletes, and go to the same school as five co-athletes.

In Table 1, we report the measures of four social network characteristics [53,60] (Borgatti, Everett, and Johnson, 2013; Wasserman and Faust, 1994) for the two sports networks. *Ties* shows how many relations there are in the networks, *average degree* shows how many relations each member has on average, *density* reports the proportion of realized relations of all possible relations in a network, and *centrality* provides a measure of how evenly the social relations in a network are distributed. Finally, we report the overlap between the two sports networks and each of the non-sports networks; for example, an overlap of 0.47 between strong sports networks and school networks means that 47% of those in the strong sports networks also go to school together.

For the first question on what the social relations in sports groups look like, the answer is that these networks are diverse when inspecting the most common social network measures: in the number of ties, average degree, density, and centralization. For athletes with strong social relations to their co-athletes, the groups contain 24 of such strong ties on average, varying from 66 at most, to five at a minimum. With an average number of 13 members in the group and an average of 24 strong ties in each group, each member has about two strong social relations (average degree). The strong sports networks are not dense (0.16) and not very centralized (0.2). This indicates that these strong sports relations are rare, exclusive, and evenly distributed. As such, these strong relations could be interpreted as a type of ‘core ties’ [32].

For the weak sports network, the average number of weak relations is 78, but they vary widely from 200 to 17. With an average number of 78 weak ties per group and (the same) 13 persons on average in each group, each person on each team has about six weak social relations (average degree). The weak sports networks are (reasonably enough) denser than the strong sports networks (0.51 vs. 0.16) and are more centralized (0.31 vs. 0.20). The weak social relationships are more widespread, less evenly distributed, and closer to qualifying as ‘sympathy ties’ [32].

A second finding from Table 1 is the substantial overlap between the sports networks and the non-sports networks. On average, 47% of those who are part of the strong sports networks in the groups also go to the same school. Similarly, 68% of the strong sports network members also have ties to each other in leisure networks, and 74% have ties in social media networks. For the weak networks, the numbers are somewhat lower: 45% for school, 64% for leisure, and 62% for social media.

The question becomes how relations outside sports (school, leisure, social media) have consequences for the social relations within sports. However, overlaps in and of themselves do not prove that what happens outside sports has consequences for what goes on inside sports. ERGM modelling can help here and shows the probability of a social tie in a network depends on a set of characteristics inherent to the network (e.g., in a dense network, the probability of having a tie is higher than in a sparse network, regardless of who one is) and, as is our interest, how the probability of sports ties depends on factors exogenous to our network (e.g., if the social networks in sports depend on athletes going to school together). We ran 57 ERGM models: analyses of 27 strong and 30 weak networks.

Instead of presenting the results of all ERGMs in 57 separate tables, we have collected the regression coefficients for each of the independent variables for each type of sports network (weak and strong) in 10 figures. The first Figure 1a–e present the effects of each of our five independent variables (in Figure 1a, this is the school network) on the probability of being part of each of the weak sports networks (controlled for other variables). Each of the dots in these figures represents the effect of the chosen variable for one specific team.

In Figure 1a–e, we find each sports team represented by a dot—the ERGM (regression) coefficient—and two grey lines indicating a confidence interval for this coefficient (±2 standard errors). The vertical dotted grey line (zero line) shows a zero effect. Dots located on the left side of the zero line indicate that a coefficient for one specific group has a negative effect, and dots on the right side of the zero line show a positive effect. For groups with standard errors not crossing this zero line, effects are statistically significant (at 0.05% level). As an example, the bottom dot in Figure 1b shows that the effect of sharing leisure activities is positive for also being part of a weak sports network in one specific team, and this effect is statistically significant because the grey lines do not cross the zero line. Table 2 reports a meta-analysis summarizing the results in Figure 1 and Figure 2: the means and standard deviations for the effects of each of the non-sports networks, and exercise durability and frequency for the weak and strong sports networks, respectively.

Starting with the effect of school networks for weak sports networks, we can see (Figure 1a) that the dots (i.e., coefficients) are close to and at both sides of the zero line, which is evidence of weak and non-systematic effects. This indicates that going to the same school is not very important for joining weak sports networks. For leisure networks (Figure 1b), there were two apparent differences compared with the effects of the school networks. First, almost all dots are to the right of the zero line, which indicates a positive and more consistent effect on weak sports networks. Eight of these coefficients are also statistically significant (at the *p* = 0.05 level), which further points to the importance of the effects of sharing leisure effects. Moving on to the social media networks (Figure 1c), we find all but one dot to the right of the zero line (i.e., positive effects), which means that being part of the same social media network is positively associated with tie development in weak sports networks. Compared with the effect of leisure networks, the dots are even farther to the right, which is indicative of larger effect sizes. Nine of these effects are also statistically significant at the *p* = 0.05 level. Comparing the weak sports networks models (Table 2), we find social media (1.20) has the largest effect as compared to leisure (1.04) and school (0.13). For the activity in the sports group itself, we first see that the effect of time spent in the clubs—length of affiliation (Figure 1d)—is small and unsystematic. The dots are close to and on both sides of the zero line, yet there are also three significant positive effects (and no negative). For exercise frequency (Figure 1e), we see that most effects are low, but there are also some positive statistically significant effects, indicating that the frequency of exercise is somewhat more important for social networks within sports than duration.

For the strong networks, the effects have similar patterns (Figure 2a–e). The school network has small and non-significant effects, and leisure and social media networks have mostly positive effects, many of them being statistically significant. The ranking of the effects is the same as that for weak network: social media has the strongest effect as compared to leisure and school (Table 2).

Our hypotheses on the effects of the frequency of contacts within sports (H1: Frequency) and the duration of contacts (H2: Affiliation) are mostly confirmed for frequency, whereas the effects of duration are less clear. For the ranking of importance of the social arenas outside sports, we assumed that leisure would have the strongest outside effect on social relations in sports (H3: Leisure). The leisure effects are strong, but not the strongest; therefore, H3 is nuanced. Next, our hypothesis that social relations in social media would have a stronger effect on social relations in sports than school (H4: Social Media) is supported: social media is more important than school networks, but the results also point to social media as carrying more weight than leisure networks for social relations in sports. What these findings imply, however, is less apparent. An important purpose of our study is to show that even though the social significance of sports is often emphasized, not all social relations in sports are necessarily deep, intimate, or committed. We have distinguished between weak and strong sports networks and assumed that strong social networks depend more on non-sports social relations than weak sports networks. This hypothesis (H5: Weak vs. strong networks) has been confirmed.

## 5. Discussion

To understand how sports provide functional social arenas for young athletes, three questions need answers: What do social relations in youth sports look like? How do these relations come about? How do these relations have consequences? The main purpose of our study—and the question least investigated so far in previous research—is the second question of what drives the development of social networks in youth sports. Our approach has been to focus on one such driver of social relations in sports: how social relations outside sports matter for the social relations within sports.

So far, we have only referred briefly to the third question about the consequences of social relations in sports, yet we will end our study with a discussion of how our results on the development of sports networks matter for one of the core outcome questions for sports scientists: How do social relations in and around sports matter for participation in sports? For grassroots sports, an obvious starting point is to assume that the social side of sports matters for participation: starting with sports, continuing with sports, and dropping out of sports.

For starting sports, it does not make too much sense to include our topic of relations between outside and inside sports. We know, however, that previous research has shown that recruitment to sports and other organizations relies less on individual characteristics than social networks: family, friends, school, and work [22,23]. Thus, it seems reasonable that social network mechanisms—contact, contagion, and homophily—also matter for how social relations outside sports influence recruitment to sports: meeting someone at school (contact), finding common ground with some new acquaintances (homophily), and being influenced, for example, by the new friend’s brother, who is already active in sports (contagion) might lead two friends to look for a sports club.

Research reports that most athletes appreciate the social aspects of sports [18,61]. Combining this well-known finding with our result—that social relations in sports depend on social relations outside sports—the present study has provided important new insights into the topic of how social networks influence sports participation, both by giving access to and using the resources embedded in the social networks [15]. It seems reasonable to assume that supporting and helping athletes with meaningful social relations outside of sports while also participating in sports increases the probability of continuing with sports. In short, to keep youth in sports, in addition to organizing high-quality sports, one should also support their social networks outside sports and perhaps do so in more than one type of non-sports network. For sports clubs, this can be done in several ways: the clubs can take the initiative for non-sports activities, they can link up with other relevant non-sports voluntary organizations, they can cooperate more closely with schools (schools and sports clubs are often in geographical and demographic proximity to each other), and they can facilitate social meeting rooms designed for group members on social media platforms.

Keeping adolescents in sports is very much the same as keeping them from leaving sports; for dropout cases, many of the same issues matter as for continuing—having a good time in sports requires vibrant social relations inside sports, and these social relations benefit from the same people being together outside sports. Furthermore, having social relations outside sports could, apart from keeping people in sports, help handle the dropout that will inevitably occur for a lot of young athletes. For many, ending sports will be a stressful experience, and having outside networks could be of help in securing a dignified exit from sports. This goes for grassroots sports [62], but it could also be worth considering for elite sports, especially for those involved in talent development schemes [63,64,65].

Taken together, the answer to recruitment to sports and the maintenance of high and enduring participation rates in sports is to emphasize social relations both inside and outside sports because they are reciprocally supportive. There could be a risk of promoting young athletes’ social lives beyond sports because attractive social relations outside sports could make sports a redundant social arena, leaving people feeling satisfied and sufficient with their non-sports social relations. Given the interplay between social relations in various arenas, this is a risk that sports officials should accept. Measures of participation in sports—durability and exercise frequency– were primarily included as control variables, but the frequency of participation has a particular effect that reminds us of the fact that the quantity and quality of sports participation are important to realize the social potential laid out in the philosophy of sports. In summary: participating in more social arenas could be individually satisfying and organizationally useful.

## 6. Conclusions

In Norway, 93% of youth take part in sports for longer or shorter periods, and a primary reason for doing so is the social outcomes of sports: meeting and making friends. Knowledge of the social aspects of youth sports then becomes pivotal. We touched on three questions, described the social structures of youth sports, and discussed some of the implications of (good) social relations for participation in youth sports. Our main question was the most neglected of the three questions: How do social relations in sports develop? Our answer to this question focused on how social relations in non-sports activities matter for the social relations in sports.

Social relations between athletes are diverse, and as a start, we distinguished between those having weak and strong relations with their co-athletes. We studied these relations considering athletes’ social relations outside sports: whether they go to the same school, whether they share one or more leisure activities, and whether they are together on social media.

We assumed that there are forces inherent in sports and the way sports are organized in voluntary organizations that support establishing social relations in sports. When further studying how social relations in sports develop, we depended on three social mechanisms common in social network studies: contact, contagion, and homophily. We also considered the voluntariness and exclusiveness of school, leisure, and social media as mechanisms that would influence social relations in sports.

Based on the contact mechanism, we hypothesized—mostly as control variables—that exercise frequency and duration of sports participation would improve the social relations in sports. Exercise frequency seemed to matter for social relations, but less so for duration. We further interpreted the effects of school, leisure, and social media relations in light of the five social mechanisms and (although a bit exploratively) assumed that all non-sports participation should matter for social relations in sports, but leisure more so than school, and probably also more than social media. The results did not fully support these hypotheses: social media seems to be the most influential as compared to leisure and school. Because strong social relations in sports are more demanding than weak social relations, our last hypothesis stated that non-sports relations are more consequential for strong than weak networks, and this assumption was confirmed.

Our study is among the first to explore how social relations in sports develop, and there are many crucial and interesting questions that need future research. Previous research has shown gender differences when it comes to social networks in general [32] and in sports [66]. A first challenge then is to adopt a gender perspective and go deeper into the question of how social relations develop within the context of sports for boys and girls. A second challenge is to develop a more nuanced network typology. As usual in network studies, we worked with a relatively simple distinction (weak and strong) between social network types in sports. It could be useful to work with more fine-grained typologies when looking at sports relations in light of the differences between gender, and also differences between sports, age, competitive levels, and organizational forms. A further challenge is to understand the social mechanisms operating in sports. These challenges also point towards the usefulness of more qualitative approaches that could dig deeper into the inherent content and meaning of social relations in sports. Future work should also seek to address a more nuanced understanding of nonsporting arenas: schools are more diverse than our data allow for, non-sports leisure activities are diverse, and we have merged them into one overall category. Theoretically, social media is a moving target and could be operationalized in many ways, and an overall question is (still) about the meaning of social media: Does social media simply reflect real-world networks, or do they represent more genuine social forces of their own [32]? We see a set of methodological challenges, and for social network studies in particular, one stands out. Our data did not allow for more than degree as an endogenous variable, yet future studies should provide data (or apply methods) that take better care of the genuine network character of the social relations in sports.

## Figures and Tables

**Figure 1 ijerph-18-06197-f001:**
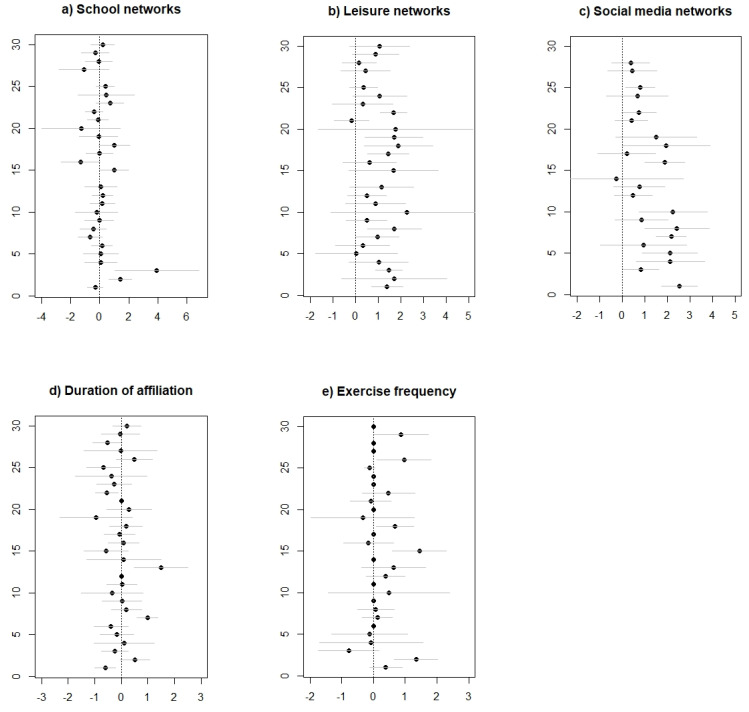
(**a**–**e**): Coefficients and standard errors (±2) for each of the independent variables in the weak network models.

**Figure 2 ijerph-18-06197-f002:**
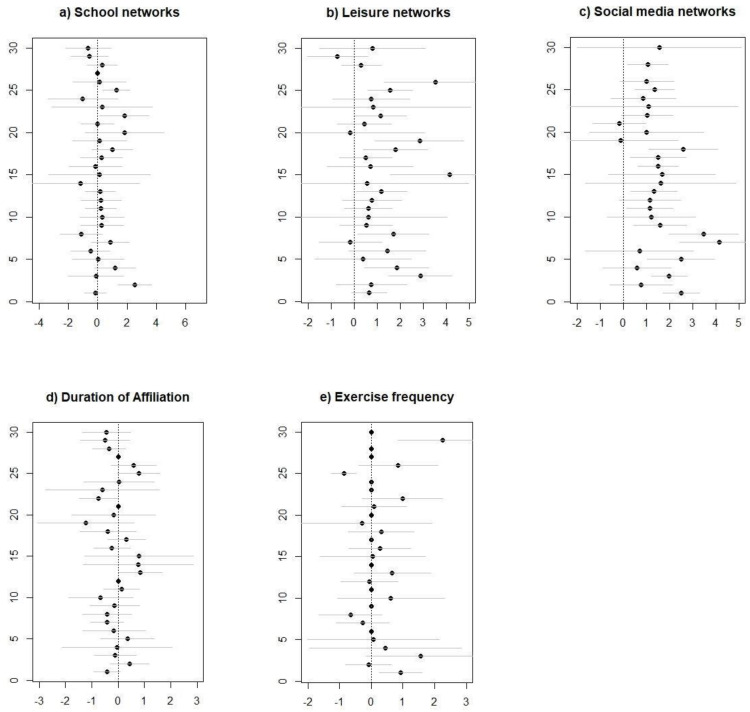
(**a**–**e**): Coefficients and standard errors (±2) for each of the independent variables in the strong network models.

**Table 1 ijerph-18-06197-t001:** Descriptive statistics of (i) network ties and (ii) proportion of overlap between networks.

	Strong Sport Networks					
	Range	Mean	Max	Min	SD	N
Size of teams	6:20	13	20	6	3.57	27
Ties per team	5:66	24	66	5	15.1	
Average degree	0.4:5.1	1.9	5.1	0.4	1.17	
Density	0.06:0.42	0.16	0.42	0.06	0.07	
Centralization	0.1:0.5	0.2	0.5	0.1	0.09	
Overlap with School networks	Ratio: 0:1	0.47	0.86	0	0.20	27
Overlap with Leisure networks	Ratio: 0:1	0.68	1	0.33	0.15	27
Overlap with Social media networks	Ratio: 0:1	0.74	1	0.40	0.18	27
	**Weak Sport Networks**					
	Range	Mean	Max	Min	SD	N
Size of teams	6:20	13	20	6	3.39	30
Ties per team	17:200	78	200	17	40.7	
Average degree	1.3:15.5	6.1	15.5	1.3	3.16	
Density	0.15:0.78	0.51	0.78	0.15	0.16	
Centralization	0.17:0.44	0.31	0.44	0.17	0.07	
Overlap with School networks	Ratio 0:1	0.45	1	0	0.26	30
Overlap with Leisure networks	Ratio 0:1	0.64	1	0.20	0.25	30
Overlap with Social media networks	Ratio 0:1	0.62	1	0.33	0.20	30

Note: M = Mean number of ties in network. Max = Maximum number of ties in network. Min = Minimum number of ties in network value. SD = Standard Deviation. N = Sample size: total number of sport teams.

**Table 2 ijerph-18-06197-t002:** Average values for the ERGM coefficients and their standard deviation in the two sport networks.

	Weak Sport Networks		Strong Sport Networks	
	Mean	SD	Mean	SD
School Networks	0.13	0.57	0.25	0.82
Leisure Networks	1.04	0.61	1.11	0.87
Social Media Networks	1.20	0.46	1.46	0.77
Duration of affiliation	−0.04	0.36	−0.07	0.50
Exercise frequency	0.20	0.28	0.23	0.41

## Data Availability

The data presented in this study are available on request from the corresponding author.

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
