# Peer review of "Friends in Sports: Social Networks in Leisure, School and Social Media"

_ijerph, 2021, doi:10.3390/ijerph18126197_

Round 1
Reviewer 1 Report
Related to the reflections made by the authors, I would like to recommend the incorporation of gender as a transversal element in social relations, which may help to understand them better.
I would like to suggest some articles whose review could enrich the critical apparatus of the article, especially in the analysis of the results and the final reflections. Some lines of research on the development of social relations may be enunciated if the gender perspective is applied.
- https://www.redalyc.org/articulo.oa?id=54252697010
- http://editorial.upnvirtual.edu.mx/index.php/publicaciones/descargas/category/1-pdf?download=423:a-look-into-masculine-identity
Author Response
Comment 1:
Related to the reflections made by the authors, I would like to recommend the incorporation of gender as a transversal element in social relations, which may help to understand them better.
Response 1:
We share your view that gender probably plays a role in the development of social relationships. That said, there are especially three factors that make it difficult to incorporate gender in the analysis of this article. First, including gender as suggested involves changing the article to something other than what it is now. Second, adding and placing more focus on gender will cause the manuscript to be too long. Given the manuscript's format, content, and deadline for resubmitting a new draft of the manuscript, there is unfortunately no room or time to make such a change at this time. Third, to adequately address gender, a larger sample size (more groups) would have been desirable.
Comment 2:
I would like to suggest some articles whose review could enrich the critical apparatus of the article, especially in the analysis of the results and the final reflections. Some lines of research on the development of social relations may be enunciated if the gender perspective is applied.
Response 2:
We have revised our section on future research and placed more emphasis on gender as a future research topic in the study of the development of social relations in sport (See lines 596-598).
Reviewer 2 Report
After reviewing the article thoroughly, I consider that the work you provide is relevant and I congratulate you. Also, I consider that a small revision of the writing in English would be necessary because I have detected small errors, otherwise, research, method, etc, it is a good job. The minor changes I refer to are a brief English revision of the article's content.
Author Response
Comment 1:
After reviewing the article thoroughly, I consider that the work you provide is relevant and I congratulate you. Also, I consider that a small revision of the writing in English would be necessary because I have detected small errors, otherwise, research, method, etc, it is a good job. The minor changes I refer to are a brief English revision of the article's content.
Response 1:
Thank you for your review of the article, we greatly appreciate having our work evaluated. The manuscript has already been reviewed by a professional proofreader with English as the first language, but you are right in that there were minor errors in the writing. We have corrected what we found of errors throughout the manuscript. Hope this is ok now, if not, we will appreciate a specification of additional aspects of the writing that should be corrected.
Reviewer 3 Report
line number 515: improve the presentation. There is a presentation error.
All aspects and variables discussed in the results should be addressed in a more extensive discussion.
Author Response
Comment 1:
line number 515: improve the presentation. There is a presentation error.
Response 1:
You are right that this was a presentation error. This has been corrected.
Comment 2:
All aspects and variables discussed in the results should be addressed in a more extensive discussion.
Response 2:
Thank you for pointing this out. You are right in that we mostly discuss the significance of the nonsport networks (school, leisure and social media) and how the social ties at these arenas are connected to and matter for social ties within sports, and to a lesser extent discuss the significance of the results on the control-variables. To account for this, we have, accordingly, highlighted and placed more emphasis on the results pertaining to durability and exercise frequency of sport participation (lines 556-560) in the discussion. We also added a sentence about social media (lines 538-539) in the discussion.